# A Review of *Dendrophthoe pentandra* (Mistletoe): Phytomorphology, Extraction Techniques, Phytochemicals, and Biological Activities

Mohd Azrie Awang [1,2,*], Nik Nurul Najihah Nik Mat Daud [3], Nurul Izzati Mohd Ismail [4], Farah Izana Abdullah [5] and Mohammad Amil Zulhilmi Benjamin [6]

1   Faculty of Food Science and Nutrition, Universiti Malaysia Sabah, Jalan UMS, Kota Kinabalu 88400, Sabah, Malaysia
2   Innovative Food Processing and Ingredient Research Group, Faculty of Food Science and Nutrition, Universiti Malaysia Sabah, Jalan UMS, Kota Kinabalu 88400, Sabah, Malaysia
3   Faculty of Medicine, Universiti Sultan Zainal Abidin Kampus Perubatan, Kuala Terengganu 20400, Terengganu Darul Iman, Malaysia
4   Department of Science and Mathematics, Centre for Diploma Studies, Universiti Tun Hussein Onn Malaysia, Hab Pendidikan Tinggi Pagoh, KM 1, Jalan Panchor, Panchor 84600, Johor, Malaysia
5   International Institute of Aquaculture and Aquatic Sciences (I-AQUAS), Universiti Putra Malaysia, Lot 960 Jln Kemang 6, Port Dickson 71050, Negeri Sembilan, Malaysia
6   Borneo Research on Algesia, Inflammation and Neurodegeneration (BRAIN) Group, Faculty of Medicine and Health Sciences, Universiti Malaysia Sabah, Jalan UMS, Kota Kinabalu 88400, Sabah, Malaysia
*   Correspondence: ma.awang@ums.edu.my

**Abstract:** *Dendrophtoe pentandra*, popularly known as mistletoe, is a semi-parasitic plant that can grow on many host plants and is native to Southeast Asia, especially Malaysia. *D. pentandra* is regarded as an unfavourable plant for an economically significant horticultural plant; however, *D. pentandra* is also regarded as a medicinal plant and is used in traditional and alternative medicine to treat coughs, diabetes, hypertension, and cancer. Scientific research has also found that this plant has significant potential for medical properties such as antioxidant, antibacterial, anticancer, antiproliferative, antidiabetic and antihyperglycaemic, anti-inflammatory, cytotoxicity, hepatoprotective, immunomodulatory, and anti-aging properties. Numerous phytochemical compounds from *D. pentandra* extracts have been identified using extraction techniques such as maceration and reflux. Thus, this review aims to provide thorough information on phytomorphology, extraction techniques, phytochemicals, biological activities, and the future prospects of *D. pentandra*.

**Keywords:** *Dendrophtoe pentandra*; phytomorphology; extractions; phytochemicals; biological activities

## 1. Introduction

Mistletoe, scientifically known as *Dendrophtoe pentandra*, is a semi-parasitic plant that can grow on many species of host plants. This plant has been called different names depending on the host plant it has grown on [1]. *D. pentandra* commonly grows in tropical rainforests at lowland plantations covering India to Indochina, Peninsular Malaysia, Indonesia (Sumatra, Jawa, Kalimantan, and Nusa Tenggara), and the Philippines [2]. Physically, *D. pentandra* is an unwanted semi-parasitic plant with branches of the woody shrub that can attain a height from 0.5 up to a maximum of 2 m tall. Although *D. pentandra* is a semi-parasitic plant, its bioactivities may also depend on its host plant [3]. *D. pentandra* has been reported to have a high potential for medicinal values such as antioxidants [1,4–10], antibacterial [6,7], anticancer, antiproliferative [8,11–18], antidiabetic, antihyperglycaemic [1,19], anti-inflammatory [20], cytotoxicity [1,14,16], hepatoprotective [19], immunomodulatory [21], and anti-aging [22] activities. The phytochemical analysis of *D. pentandra's* leaves and flowers also revealed the presence of alkaloids, phenolics, flavonoids, saponins, and terpenoids [5,10,16,23].

Different extraction procedures can be used to extract phytochemicals from medicinal plants. They generally have two sorts of extraction procedures: traditional and modern [24]. Traditional extraction procedures, such as maceration, reflux, and Soxhlet extraction, necessitate a large amount of solvent and a lengthy extraction duration, resulting in a labour-intensive, time-consuming, and environmentally unfriendly process. By contrast, modern extraction techniques such as microwave-assisted extraction (MAE), accelerated solvent extraction (ASE), supercritical fluid extraction (SFE), and ultrasonic-assisted extraction (UAE) have many advantages, such as reducing the usage of organic solvents, improving extraction efficiency and times, and minimising bioactive compound degradation in the sample [24]. Hence, this review aims to study *D. pentandra* in detail, focusing on phyto-morphology, extraction techniques, phytochemicals, and biological activities. These details could help researchers and scientists learn more about this plant in the future.

## 2. Phytomorphology

Generally, the phytomorphology of *D. pentandra* consists of leaves, fruits, roots, and flowers, and each part of the plant can be used for medications [2,25]. Figure 1 shows how the *D. pentandra* plant attaches itself to a host plant.

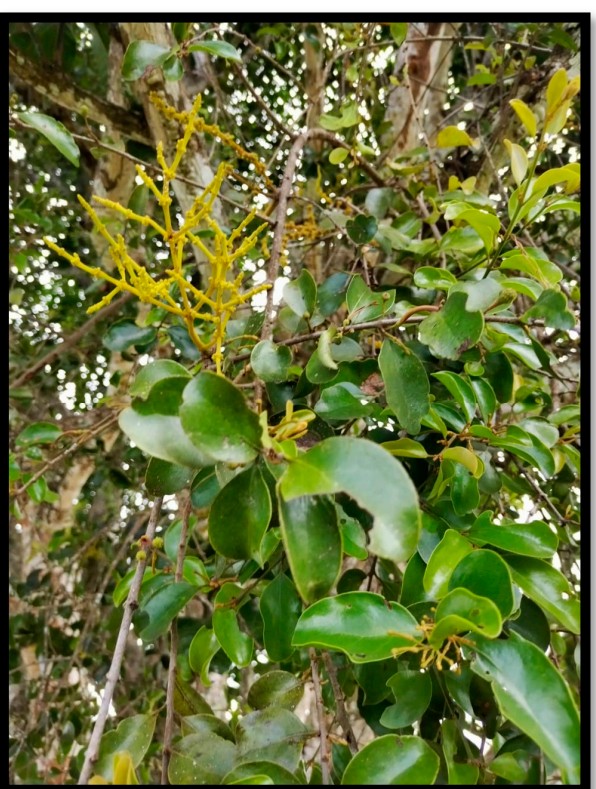

**Figure 1.** *D. pentandra* plant attaches itself to a host plant.

### 2.1. Leaves

*D. pentandra's* leaves have simple features. They are green-coloured and arranged alternately along the branches [25]. The leaves are narrow, oval shaped, also known as lanceolate, close to an elliptic or sub-orbicular shaped 8–10 cm long and 5–15 cm wide [2,25]. The petioles are 5–20 mm long. The features of the young shoots are covered with pubescent (tiny hairs), and the young leaves are purplish-red [25]. The mature leaves are green, with thick, leathery leaf blades and a glossy surface [2,25]. The leaves have 2–4 pairs of lateral veins, base cuneate or obtuse, apex acute or rounded, and glabrous [2]. The branches or stems are greyish and brownish and scattered lenticellate [2].

## 2.2. Flowers

The flowers of *D. pentandra* are cylindrical and are categorised as bisexual flowers [2]. The flowers are borne on axillary racemes that are solitary [2]. Each flower is about 1.5–2 cm long [2,25]. The flowers develop from orangish buds into flowers with reddish and greyish or white stellate hair [25]. The flowers then bloom with five narrow petals folded back, showing an orange interior and greenish exterior [25]. The corolla (petals of the flower) is orange and slightly inflated at the base, while the corolla tube is hairy, and the stamen stalks and style are greenish and extruded [25]. The fully developed flowers usually form 3–10 flowers per cluster [25]. Each cluster can occur individually or in pairs or trios [25].

## 2.3. Fruits and Seeds

The fruits of *D. pentandra* are generally small and appear as fleshy red berries that are 8–10 mm long and are minutely pilose or glabrous [2]. The young fruits are yellowish-green to pink, while the ripened ones are reddish [2]. *D. pentandra* fruits come from a single-seeded berry: a drupe without a stony endocarp. The enclosed seed is surrounded by viscin, which is composed of cellulosic strands confined by mucilaginous pectic material that adheres the seed to the host plant following dispersal [26]. The seeds have a viscous coating that adheres to any substrate. *D. pentandra* seeds are dispersed by birds that consume *D. pentandra* fruits [27,28].

## 2.4. Roots

As a semi-parasitic plant, the stem of *D. pentandra* has designed roots known as haustoria which are pricked into the host tissues and connected to the xylem or phloem as the host [26]. The haustoria look similar to a ball that is attached to a branch. The penetration of the root gives a water supply and is non-organic from the host [27]. *D. pentandra* attacked the host tree in the branch part, and then at the point of the branch, which was infected by *D. pentandra*, there was a swelling form called haustoria. From the closest part of the branch, nutrients, and photosynthate migrated primarily to *D. pentandra* as opposed to the distal portion. It obstructed the distal part of the branch's growth, and then the distal part of the branch gradually died [28].

Almost all the parts of the *D. pentandra* have been used traditionally for traditional medicinal and alternative medicinal purposes. For example, the extract from *D. pentandra* leaves has been used for diabetic and aging-related diseases due to its antidiabetic [1,19] and antioxidant properties [1,4–10]. Moreover, *D. pentandra* flowers, when extracted, have also been studied due to their antioxidant properties [5]. Interestingly, Pramestya et al. [29] found that *D. pentandra* flowers tend to accumulate more secondary metabolites than other parts. The extract shows high antioxidant activities, which marks the potential for aging and skin-related diseases. The stem and root phenol extracts contain antioxidant properties that inhibit free radicals and cause degenerative diseases such as cancer, heart, and diabetes [30].

## 3. Extraction Techniques

Extraction is a method that allows components or compounds to be obtained from samples (plants). There are two sorts of extraction procedures: traditional and modern (Figure 2). Several extraction methods have been proposed so far, e.g., maceration, reflux, Soxhlet extraction, MAE, ASE, SFE, and UAE [31,32]. These chosen methods were applied based on the type of compounds that needed to be extracted, the properties of the compounds and raw materials, and the cost of extraction. The chosen extraction process also depends on other factors, including rapidness, reliability, and environmental friendliness.

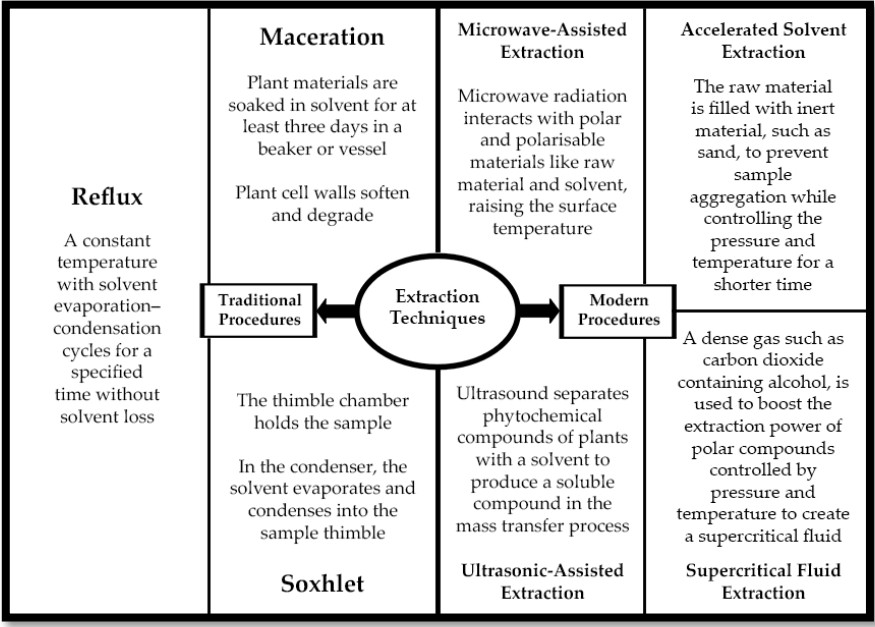

**Figure 2.** Pictorial representation of traditional and modern procedures of extraction techniques.

### 3.1. Maceration

Maceration is the most common and straightforward extraction technique for medicinal plants. This technique involves soaking plant materials in a beaker or vessel with a solvent at room temperature for a minimum of three days or up to a few weeks, depending on the characteristic of the plant and solvent used with frequent agitation [33]. In order to extract the phytochemicals from the raw material, this procedure softens and degrades the cell wall of a plant.

### 3.2. Reflux

This reflux extraction method is widely used to extract phytochemicals from medicinal plants. The extracted sample could weigh up to a few kilograms. Because the solvent is constantly heated, the phytochemicals in the plant may be destroyed, especially heat-sensitive ones. It is incompatible with the extraction of thermolabile natural compounds. Nevertheless, this method is more effective than maceration or percolation because it requires less solvent and extraction time. This technique can be conducted at a constant temperature with repeated solvent evaporation–condensation phases for a specified period of time without solvent loss [34]. This technology is believed to be widely used in the herbal sector since it is efficient, simple to use, and inexpensive [35].

### 3.3. Soxhlet

Soxhlet extraction is one of the most common procedures for extracting herbal plants. In this process, a raw material of small size is deposited in the Soxhlet apparatus's thimble chamber in a thimble composed of cellulose or strong filter paper. The extraction solvent is heated from the flask's bottom. The solvent vaporises, condenses in the condenser, and drops into the sample thimble [36,37]. This approach, however, takes longer than the reflux method since it requires at least eight hours for complete extraction [38]. The same solvents (methanol, ethanol, ethyl acetate, and acetone) can be used in Soxhlet extraction as well as in maceration or reflux extraction. The solvent is recycled and is not released into the environment. At the end of the process, the extract can be concentrated (extractum spissum) or evaporated (extractum siccum), but the solvent may be collected and reused.

### 3.4. Microwave-Assisted Extraction

MAE utilises microwave energy to transfer solutes from a sample matrix (raw material) to a solvent medium [39]. The radiation from the microwave interacts with the dipoles of polar and polarisable materials such as the solvent and raw material, thus elevating the surface temperature. According to Kaufmann et al. [40], the dipole rotation of molecules induced by the microwave electromagnetic field disrupted the hydrogen bonding, thereby facilitating the migration of dissolved ions and promoting solvent penetration into the matrix of samples. This technique can hasten the conventional extraction processes, such as Soxhlet and maceration techniques. Nonetheless, this method is limited to small-molecule phenolic compounds such as phenolic acids (e.g., gallic acid and ellagic acid), quercetin, isoflavone, and trans-resveratrol because these molecules were stable up to 100 °C for 20 min [41].

### 3.5. Accelerated Solvent Extraction

ASE is relatively new and is more efficient than conventional methods such as maceration and Soxhlet methods, as it uses less solvent than other techniques. In the extraction cell made of stainless steel, the raw material or sample of medicinal plants is filled with inert material, such as sand, to prevent sample aggregation [42]. ASE can control the pressure and temperature for different samples, and its operating time is usually shorter than other techniques [42]. The working principle of ASE is similar to those of other solvent techniques. ASE was also suggested for the quality control evaluation [43]. This proposed approach also provided the base for productive green extraction [44].

### 3.6. Supercritical Fluid Extraction

Another modern extraction technique is SFE. Dense gas is employed, which is a substance with the physical properties of both a gas and a liquid at its critical point. Carbon dioxide ($CO_2$) is the most often used solvent since it is non-toxic, non-flammable, and non-viscous. Historically, SFE was employed to extract essential oils or less polar components from plants. Because the $CO_2$ utilised in this technique had low solubility for polar compounds, it could not successfully extract the polar phytochemicals in a sample. Alcohol has been occasionally used as an addition to increase the extraction power of polar compounds. SFE is affected by the fluid solvent's viscosity, density, diffusivity, and dielectric constant [45]. In-depth, pressure and temperature are two crucial parameters in SFE to achieve a supercritical fluid. By modifying the pressure, temperature, and co-solvent concentrations, SFE can adjust the solvent power for various compounds [46]. Furthermore, the initial cost of the equipment should be highly considered [47].

### 3.7. Ultrasonic-Assisted Extraction

UAE is a type of solid–liquid extraction. It involves the mass transfer process that separates the phytochemical compounds of plants with the help of a suitable solvent, where the soluble compound is the target product. This method uses an ultrasound ranging from 20 to 2000 kHz [33]. The resulting acoustic cavitation increases the surface contact between the solvent and the raw material and the permeability of the cell walls [48]. The advantages of this technique include a higher yield, lower solvent consumption, shorter extraction time, an environmentally friendly process, better solvent penetration, increased mass transfer, and lower capital investment [49]. There are two types of ultrasonic equipment that have been used to assess the ability of sonication in extracting phytochemicals: ultrasonic baths and probes. The exploration of the usage of ultrasonic probes for the extraction of phytochemicals was started with the ultrasonic bath. Nevertheless, the ultrasonic probe has the advantage of shorter time extraction as it is more powerful and efficient than the ultrasonic bath [50]. The ultrasonic probe is also much better at delivering intensity towards smaller surfaces than the ultrasonic bath. Another reason for this is that the ultrasonic bath has high attenuation because of the water contained bath and the walls of the flask

(indirect). By contrast, for the ultrasonic probe, attenuation was less because the probe was immersed directly into the flask [50].

All extraction techniques were used to extract the *D. pentandra* plant, as shown in Table 1. Based on the previous studies, almost all extraction techniques used maceration for the *D. pentandra* extracts. In summary, traditional techniques such as maceration and reflux were still practical for the *D. pentandra* extract investigation.

**Table 1.** Summary of extraction techniques of *D. pentandra* extracts.

| Host Plant | Extraction Technique | Solvents Used | Targeted Compound | Yield (%) | Ref. |
|---|---|---|---|---|---|
| *Annona squamosa* *Camellia sinensis* *Spondias dulcis* *Stelechocarpus burahol* | Maceration | M, A | – | – | [1] |
| *Averrhoa carambola* | Maceration | E | Quercitrin and flavanol glycoside | – | [4] |
| *Bauhinia purpurea* | Maceration | M | – | – | [5] |
| Duku | Maceration | M, H, EA | – | – | [6] |
| *Lansium domesticum* | Maceration | M, H, EA | Quercitrin | – | [7] |
| – | Maceration | E | – | – | [8] |
| *Bauhinia purpurea* *Mangifera indica* *Stelechocarpus burahol* | Maceration | M | – | 27.1 19.6 29.5 | [9] |
| *Moringa oleifera* | Maceration | M, A M, EA M, H | – | 12.2 10.2 26.7 | [10] |
| – | Maceration | M | – | 35.8 | [11] |
| Mango | Maceration | E | Quercetin | – | [12] |
| – | Maceration | M | – | – | [13] |
| Clove | Reflux Maceration | A E | – | – | [14] |
| – | Maceration | C DE EA M PE | – | – | [15] |
| *Lansium parasiticum* | Maceration | C DE EA PE | – | 70.0 12.5 5.0 12.5 | [16] |
| Mango | Maceration | E | – | – | [17] |
| Mango | Maceration | E | Quercetin | – | [18] |
| – | Maceration | M | – | – | [19] |
| Mango | Maceration | E | Quercetin | – | [20] |
| – | Maceration | E | – | 8.7 | [21] |
| Mango | Maceration | E | – | – | [22] |
| Mango | Maceration | E | – | 11.0–16.0 | [23] |
| *Bauhinia purpurea* | Maceration | M | – | 10.7–33.1 | [29] |
| *Lansium domesticum* | Maceration | M | Progesterone | – | [51] |

Solvents: A—Aqueous. C—Chloroform. DE—Diethyl ether. E—Ethanol. EA—Ethyl Acetate. H—Hexane. M—Methanol. PE—Petroleum Ether.

## 4. Phytochemical Constituents

The chemical composition of *D. pentandra* has been highlighted. However, its specific medicinal uses and phytochemical properties are not described much. To date, no study in the literature has been discussed on this plant properly. Phytochemicals are the chemicals produced by plants. Nonetheless, few studies in the literature have indicated the presence of natural antioxidant compounds in *D. pentandra*, including phenolics, tannins, flavonoids, alkaloids, terpenoids, and saponins in various plant parts [5]. *D. pentandra* contains different types of biologically active compounds such as amino acids, fats, carbohydrates, oligosaccharides, polysaccharides, enzymes, flavonoids, glycoprotein (lectin MLT), polypeptide (viscotoxin), vesicles, and triterpene acids [16]. Lectins (ML-I, ML-II, and ML-III) are the primary components of *D. pentandra* and are responsible for its anticancer and immunomodulatory effects [16].

### 4.1. Phenolics and Polyphenolics

Phenolic compounds are secondary metabolites and are generated by the phenyl-propanoid metabolism of plant shikimic acid and pentose phosphate [52]. Their structure consists of an aromatic ring containing one or more hydroxyl substituents [53] (Figure 3). According to Oboh et al. [54], polyphenol plant extracts possess potential key enzymes of hypertension disease (angiotensin-converting enzyme) and type-2 diabetes mellitus ($\alpha$-amylase and $\alpha$-glucosidase). Apart from this, the polyphenolic compound possesses a myriad of vital pharmacological effects such as antioxidant, anti-inflammatory, antiviral, and anticarcinogenic [16]. Depending on the plant species and extraction solvent, the accumulation of phenolic compounds in plant parts can vary [5,23].

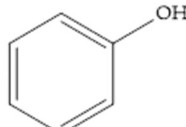

**Figure 3.** Basic structure of phenolics.

### 4.2. Tannins

In 1796, Seguin used the name "tannins" to describe the chemicals found in plant extracts that interact with proteins from animal skins to form a complex that may be converted to leather and prevent skin putrefaction [55]. Currently, tannins refer to a group of phenolic secondary metabolites of plants that can bind to proteins [56] and have a high molar mass spanning from 300 Da to 3000 Da, including those even as high as 30,000 Da [57]. There are two types of tannins, hydrolysable tannins and condensed tannins [58], as shown in Figure 4. According to Demarque et al. [55], condensed tannins are two or more units of catechin derivatives that polymerise and form complex structures. Hydrolysable tannins, on the other hand, result from the esterification of sugar with gallic acid units. In folk medicine, plants with condensed tannins treat earache, painful haemorrhoids, wounds, and swelling [58]. On the other hand, hydrolysable tannins can be used to treat faecal pathogenic bacteria [59].

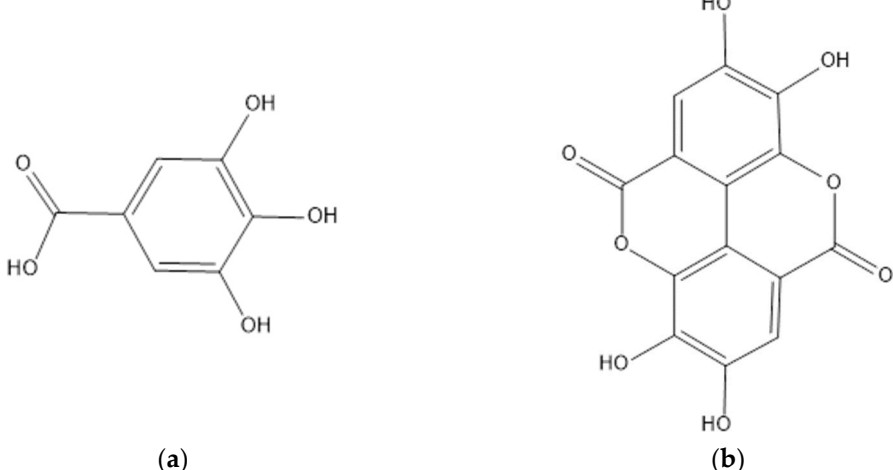

(**a**)　　　　　　　　　　　　　　(**b**)

**Figure 4.** Types of tannins and their basic structure: (**a**) Hydrolysable tannins (gallic acid); (**b**) Condensed tannins (ellagic acid).

### 4.3. Flavonoids

Flavonoids are an important class of plant secondary metabolites and are composed of a polyphenolic structure group (Figure 5). Flavonoids possess many vital pharmacological effects, and their effects depend on structural class, hydroxylation degree, other substitutions, conjugations, and polymerisation degree [60]. Flavonoids are indispensable for various medicinal, nutraceutical, pharmacological, and cosmetic applications due to

their extensive array of health-promoting properties [61]. According to Artanti et al. [4], *D. pentandra* contains a major group of flavonoids that contribute to its antioxidant activity. However, several studies also reported the pharmacological effect of flavonoids from *D. pentandra*, such as antioxidants [7], antibacterial [7], anticancer [12,18], and anti-inflammatory [20] activities. Currently, only certain flavonoids have been detected in this plant, such as quercitrin, quercetin, and flavonol glycoside [4,62] (Figure 6). In addition, quercitrin and quercetin were detected as the major compounds in a fermented *D. pentandra* leaf [62]. Nonetheless, further investigation is still required regardless of the host tree, even though it was suggested that the chemical constituents of the commercial *D. pentandra* extracts depended on the host tree [9,23].

**Figure 5.** Basic structure of flavone backbone (2-phenyl-1,4-benzopyrone).

(**a**)

(**b**)

(**c**)

**Figure 6.** Chemical structures of flavonoids found in *D. pentandra*: (**a**) Quercitrin; (**b**) Quercetin; (**c**) Flavonol glycoside.

## 4.4. Alkaloids

Alkaloids are naturally occurring compounds containing oxygen, carbon, hydrogen, and typically nitrogen, and they are found primarily in plants, particularly certain floral plants. Previously, there have been several studies conducted on phytochemical screenings in *D. pentandra*, and these results show that there is a presence of alkaloid content in *D. pentandra* [5,10,16,23]. Phytomedicines that are derived from plants, such as alkaloids,

have been used against neurodegenerative diseases since antiquity [63]. Many alkaloids are valuable medicinal agents that can treat diseases such as malaria, diabetes, cancer, and cardiac dysfunction [64]. Generally, there are two broad divisions of alkaloids, including heterocyclic (typical alkaloids) and non-heterocyclic (atypical alkaloids) [65]. Nevertheless, alkaloids are typically categorised based on the nature of the fundamental chemical structures from which they are derived into a number of categories that include imidazole, indole, isoquinoline, piperidine, purine, pyrrolizidine, quinolizidine, tropane, and pyrrolidine alkaloids [66]. In short, five major alkaloid subgroups were identified, namely free alkaloids (H-heterocycles), protoalkaloids (N-containing side), polyamine alkaloids, cyclopeptide alkaloids, and pseudoalkaloids, with representative alkaloids, which are shown (Figures 7–11) [67]. In 1986, Khwaja et al. [68] reported that the presence of alkaloids in the *Viscum album* (European mistletoe) extract contributed to the antitumor properties of the plant against colon adenocarcinoma 38, Lewis lung carcinoma, and C3H mammary adenocarcinoma 16/C. This breakthrough has recently brought a new realisation to therapeutic studies involving this semi-parasitic plant, and a few controlled clinical trials have been started. Previously, the *V. album* extract has been suggested for use in the postoperative treatment of lung, breast, and gastrointestinal cancers, indicating the potential of *D. pentandra* in treating cancer [69].

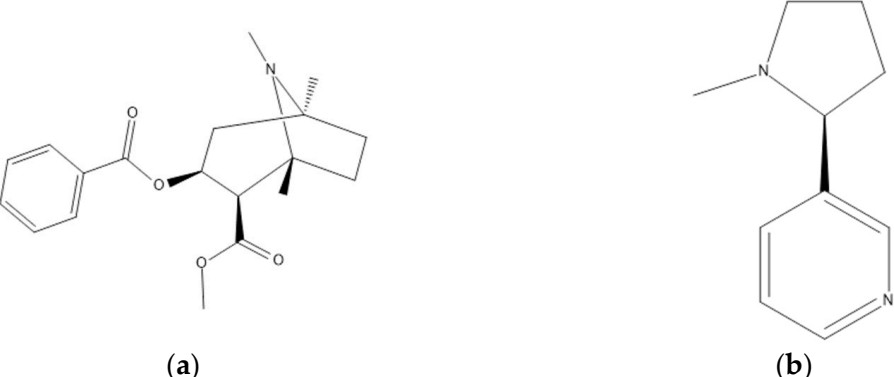

(**a**)　　　　　　　　　　　　　　　　(**b**)

**Figure 7.** Example of chemical structures of free alkaloids (H-heterocycles): (**a**) Cocaine; (**b**) Nicotine.

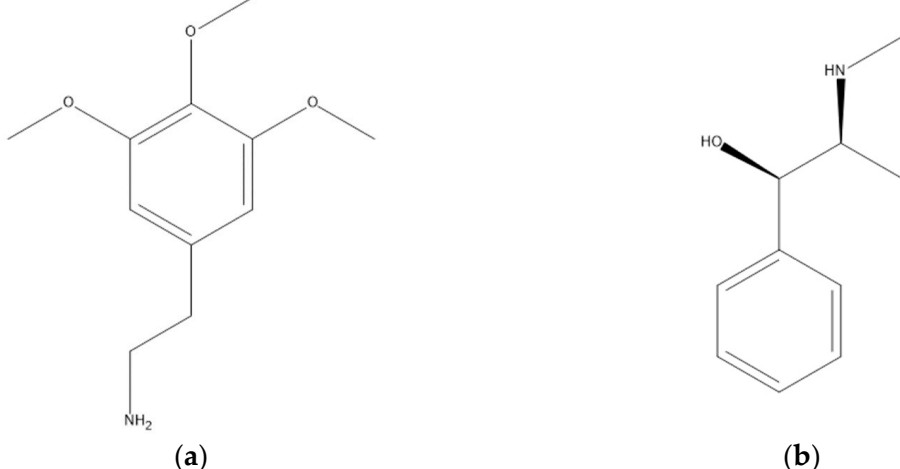

(**a**)　　　　　　　　　　　　　　　　(**b**)

**Figure 8.** Example of chemical structures of protoalkaloids (N-containing side): (**a**) Mescaline; (**b**) Ephedrine.

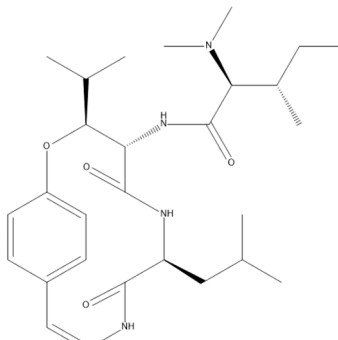

**Figure 9.** Example of chemical structures of polyamine alkaloids: (**a**) Palustrine; (**b**) Paucine.

**Figure 10.** Example of chemical structures of cyclopeptide alkaloids: Frangulanine.

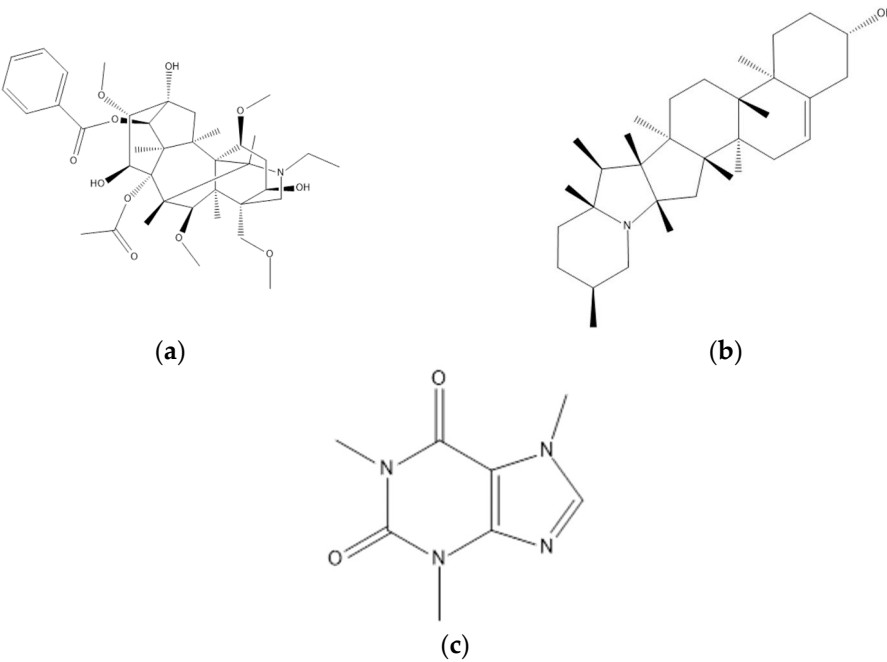

**Figure 11.** Example of chemical structures of pseudoalkaloids: (**a**) Aconitine; (**b**) Solanidine; (**c**) Caffeine.

### 4.5. Terpenoids

Terpenoids are organic compounds found in nature that are derived from isoprene units [70] (Figure 12). Several studies have shown rising colour stains in leaf extracts of *D. pentandra*, indicating the presence of terpenoids in the plant [5,10,16]. In plants, most microRNAs (miRNAs) are encoded by their primary transcript, and they regulate the expression of genes that code for transcription factors, stress response proteins, and

others that affect biological processes [71,72]. In this case, Xie et al. [73] used miRNAs to detect terpenoid biosynthesis and successfully determined 68 miRNAs target genes in *V. album*. Even though there is much information related to the miRNA regulation of specific processes, the role of miRNAs in the regulation of secondary plant product biosynthesis still needs to be better recognised.

**Figure 12.** Isoprene unit derived in terpenoids.

*4.6. Saponins*

Saponins are a type of natural product that is made up of aglycone (triterpene or steroid) and sugars (hexose or uronic acid) [74] (Figure 13). To date, saponin detection in *D. pentandra* has been conducted only involving qualitative tests, and no other test has been conducted [5,16]. According to Yee et al. [16], phytochemical screening revealed that saponins, tannins, flavonoids, and alkaloids were present in the leaves of a *D. pentandra* ethyl acetate extract, suggesting that this plant could be considered for the development of a new chemotherapeutic agent with promising potency.

**Figure 13.** Basic structure of saponins.

## 5. Biological Activities

From a biological activities aspect, *D. pentandra* extracts reported major biological activities such as antioxidant, antibacterial, anticancer, antiproliferative, antidiabetic, antihyperglycaemic, anti-inflammatory, cytotoxicity, hepatoprotective, immunomodulatory, and anti-aging activities which support their traditional usage to treat ailments. The biological activities of *D. pentandra* extracts depend on their host plants and the extraction solvent used [1,6,14,15]. Table 2 shows the summary of the biological activities of *D. pentandra* extracts.

**Table 2.** Summary of biological activities of *D. pentandra* extracts.

| Biological Activities | Host Plant | Solvents Used | Results | | +/− Control | Ref. |
|---|---|---|---|---|---|---|
| Antioxidant Activity | *Stelechocarpus burahol* | M | IC$_{50}$= | 21.5 µg/mL | − | [1] |
| | | A | | 299.0 µg/mL | | |
| | *Spondias dulcis* | M | | 30.9 µg/mL | | |
| | | A | | 445.0 µg/mL | | |
| | *Annona squamosa* | M | | 22.9 µg/mL | | |
| | | A | | 741.0 µg/mL | | |
| | *Camellia sinensis* | M | | 84.9 µg/mL | | |
| | | A | | 303.0 µg/mL | | |
| | *Averrhoa carambola* | E | IC$_{50}$= | 9.05–24.72 µg/mL | − | [4] |
| | *Bauhinia purpurea* | M | IC$_{50}$= | 6.99–13.58 µg/mL | 9.57 µg/mL | [5] |
| | Duku | M | IC$_{50}$= | 2.89–13.21 µg/mL | − | [6] |
| | *Lansium domesticum* | M | IC$_{50}$= | 3.59 ppm | 5.10 ppm | [7] |
| | – | E | IC$_{50}$= | 4.74 µg/mL | 3.24 µg/mL | [8] |
| | *Bauhinia purpurea* | M | | 15.30 µg/mL | | |
| | *Mangifera indica* | M | IC$_{50}$= | 21.50 µg/mL | 3.40 µg/mL | [9] |
| | *Stelechocarpus burahol* | M | | 10.33 µg/mL | | |
| | | A | | 29.46 µg/mL | | |
| | *Moringa oleifera* | EA | IC$_{50}$= | 7.08 µg/mL | 3.46 µg/mL | [10] |
| | | H | | 10.90 µg/mL | | |
| Antibacterial Activity | Duku | M | ZOI= | 8.13–9.12 mm | − | [6] |
| | | H | | 7.85–8.97 mm | | |
| | | EA | | 7.75–9.00 mm | | |
| | *Lansium domesticum* | M, H, EA | ZOI= | 7.23–9.54 mm | NZ | [7] |
| Anticancer and Antiproliferative Activities | – | E | IC$_{50}$= | 728.05 µg/mL | NS | [8] |
| | – | M | IC$_{50}$= | 10.65 µg/mL (MCF-7 cell) | S | [11] |
| | Mango | E | Dose: | 250 mg/kg BW | NS | [12] |
| | – | M | IC$_{50}$= | 192–500 µg/mL | NS | [13] |
| | Clove | H | IA= | 38.69% (K562 cell) 41.50% (MCM-B2 cell) | S | [14] |
| | – | EA | IC$_{50}$= | 14.42 µg/mL | S | [15] |
| | | M | | 17.70 µg/mL | | |
| | | C | | 82.33 µg/mL | | |
| | | DE | | 101.57 µg/mL | NS | |
| | | PE | | 89.70 µg/mL | | |
| | *Lansium parasiticum* | EA | IC$_{50}$= | 4.72 µg/mL (MCF-7 cell) | S | [16] |
| | | | IC$_{50}$= | 18.12 µg/mL (L929 cell) | | |
| | – | E | Dose: | 25 & 50 µg/mL + 5 µg/mL (doxorubicin) | S | [17] |
| | Mango | E | Dose: | 50 µg/mL + 5 µg/mL (5-fluorouracil) | S | [18] |
| Antidiabetic and Antihyperglycaemic Activities | *Stelechocarpus burahol* | M | IC$_{50}$= | 31.8 µg/mL | − | [1] |
| | | A | | 29.4 µg/mL | | |
| | *Spondias dulcis* | M | | 41.2 µg/mL | | |
| | | A | | 34.1 µg/mL | | |
| | *Annona squamosa* | M | | 50.9 µg/mL | | |
| | | A | | 13.9 µg/mL | | |
| | *Camellia sinensis* | M | | 17.6 µg/mL | | |
| | | A | | 11.8 µg/mL | | |
| | – | M | Dose: | 400 mg/kg BW | S | [19] |
| Anti-inflammatory Activity | Mango | E | Dose: | 600 mg/kg BW | S | [20] |

**Table 2.** *Cont.*

| Biological Activities | Host Plant | Solvents Used | Results | | +/− Control | Ref. |
|---|---|---|---|---|---|---|
| Cytotoxicity Activity | *Stelechocarpus burahol* | M | | >1000 µg/mL | | |
| | | A | | >1000 µg/mL | | |
| | *Spondias dulcis* | M | | >1000 µg/mL | | |
| | | A | LC$_{50}$= | >1000 µg/mL | − | [1] |
| | *Annona squamosa* | M | | >1000 µg/mL | | |
| | | A | | >1000 µg/mL | | |
| | *Camellia sinensis* | M | | >1000 µg/mL | | |
| | | A | | >1000 µg/mL | | |
| | | A | | >1000 µg/mL | | |
| | | E | | >1000 µg/mL | | |
| | Clove | E | LC$_{50}$= | >1000 µg/mL | − | [14] |
| | | EA | | 649.12 µg/mL | | |
| | | H | | 55.32 µg/mL | | |
| | *Lansium parasiticum* | EA | LC$_{50}$= | >1000 ppm | NS | [16] |
| Hepatoprotective Activity | – | M | Dose: | 400 mg/kg BW | S | [19] |
| Immunomodulatory Activity | – | E | Dose: | 100 µg/mL | S | [21] |
| Anti-ageing Activity | Mango | E | Dose: | 600 mg/kg BW | S | [22] |

IA—Inhibition Activity. NS—Not Significant. NZ—No Zone of Inhibition. S—Significant. ZOI—Zone of Inhibition. Solvents: A—Aqueous. C—Chloroform. DE—Diethyl ether. E—Ethanol. EA—Ethyl Acetate. H—Hexane. M—Methanol. PE—Petroleum Ether.

*5.1. Antioxidant Activity*

Reactive oxygen species (ROS) include free radicals (e.g., superoxide anion and hydroxyl radicals) and non-free-radical species (e.g., hydrogen peroxide and single oxygen) [75]. ROS can result in cellular oxidative stress, contributing to the induction and progression of various diseases [75]. The plant is a source of natural antioxidants which can counteract ROS.

*D. pentandra* was reported to possess antioxidant activity based on previous studies [1,4–10]. Methanolic extracts of *D. pentandra*, which grew on *Annona squamosa*, *Camellia sinensis*, *Stelechocarpus burahol*, and *Spondias dulcis,* showed significant antioxidant activity compared to water extracts as they required a lower concentration to achieve the half-maximal inhibitory concentration (IC$_{50}$) of radicals. The IC$_{50}$ for methanolic extracts ranged between 21.5 and 84.9 µg/mL [1]. Furthermore, the findings of phytochemicals have also been brought to the isolation of flavonol glycoside and quercitrin (quercetin-3-O-rhamnoside) in ethanolic extracts of *D. pentandra* leaves that grow on *Averrhoa carambola* as well as an antioxidant component with the lowest IC$_{50}$ value of 9.05–24.72 µg/mL [4].

Both leaves and flowers of *D. pentandra* methanolic extracts were tested for their antioxidant activity [5]. The 2,2-diphenyl-1-picrylhydrazyl (DPPH) assay confirmed that flower extracts (6.99 µg/mL) presented a lower IC$_{50}$ compared to the leaf extract (13.58 µg/mL) [5]. Quantitative analysis revealed that the methanolic extract of flowers contained more phenolics and flavonoids than the leaf extract, which could contribute to high antioxidant activity [5]. Hardiyanti et al. [6] studied the antioxidant activity of the *D. pentandra* methanolic extract from duku, which resulted in IC$_{50}$ ranging from 2.89 to 13.21 µg/mL. Moreover, Hardiyanti et al. [7] successfully isolated quercitrin from the *D. pentandra* methanolic extract. From the DPPH assay, quercitrin possessed more potent antioxidant activity (IC$_{50}$ = 3.59 ppm) compared to ascorbic acid (IC$_{50}$ = 5.10 ppm) [7].

The methanolic extract of *D. pentandra* from *Stelechocarpus burahol* showed the highest antioxidant activities based on the lowest IC$_{50}$ (10.33 µg/mL), followed by *Bauhinia purpurea* (15.30 µg/mL) and *Mangifera indica* (21.50 µg/mL) [9]. In addition, *D. pentandra* extract also possessed the highest value of phenolic (431.6 mg GAE/100 g) and flavonoid (52.92 mg GAE/100 g) content, thus resulting in the highest value of antioxidant activities [9]. From the chromatogram profile, the *D. pentandra* extract exhibited the same metabolite composition even though they originated from different host plants [9]. The antioxidant activity of *D. pentandra* from *Moringa oleifera* was conducted [10]. The ethanolic extract of *D. pentandra*

was fractioned with ethyl acetate, hexane, and water [10]. Among these, the ethyl acetate fraction exhibited the highest antioxidant activity (IC$_{50}$ = 7.08 µg/mL) [10].

### 5.2. Antibacterial Activity

Plants are an alternative source of antimicrobials and are considered cheaper, natural, and safer than allopathic medicine, which has side effects [76]. Researchers have been inspired by the antibacterial potential of plants to develop effective antibacterial medications to combat infectious diseases. Antibacterial activity was also reported for *D. pentandra*. The methanolic, hexane, and ethyl acetate extracts and flavonoid isolation from *D. pentandra* grew on duku and were effective against *Escherichia coli*, *Salmonella typhi*, *Staphylococcus aureus*, and *Pseudomonas* sp. [6]. At 1000 µg/mL, the highest zone of inhibition (ZOI) was obtained from the flavonoid extract toward *S. aureus* with 9.92 mm of ZOI. Overall, the antibacterial activity of *D. pentandra* extract was dominated by flavonoid isolation [6]. This study continued by isolating quercitrin from *D. pentandra*, which grew on *Lansium domesticum*. At 1000 ppm of the concentration, against *E. coli*, *S. typhi*, *S. aureus*, and *Pseudomonas* sp. from quercitrin showed a ZOI of 7.74, 7.23, 9.54 and 8.52 mm, respectively [7].

### 5.3. Anticancer and Antiproliferative Activities

Several medicinal plant species and their phytochemicals have been reported to inhibit the progression and development of cancer. Plant metabolites such as alkaloids, biomolecules, flavonoids, gums, glycosides, lignans, minerals, saponins, taxanes, terpenes, vitamins, and oils play significant roles in inhibiting cancer progression via multiple mechanisms [77]. Numerous previous studies have been conducted for *D. pentandra* extracts on combating cancer cells [8,11–18].

An evaluation of the antiproliferative activity of *D. pentandra* methanolic extracts and its mechanism of action in cancer treatment found that the growth of MCF-7 cells was arrested at a G1/S phase with the help of p53 throughout the apoptosis pathway. This apoptosis was promoted by the inactivation of Bcl-2 and the activation of Bax proteins [11]. Additionally, ethanolic extracts of *D. pentandra*, which grew on mango, also possessed antiproliferative activity. These extracts could prevent proliferation by inhibiting the S phase of the cell cycle and by inducing p53 expression when administered with 250 mg/kg BW of *D. pentandra* extracts containing quercetin [12].

The study by Zamani et al. [13] highlighted the potential for the methanolic extract of *D. pentandra* leaves as an apoptosis inducer or cell cycle arrester in leukemic cells. This study developed two types of cell lines, imatinib-resistant K562R and K562, to mimic the resistance developed under prolonged therapy [13]. The *D. pentandra* extract had greater antiproliferative effects toward resistant cell lines of K562R (IC$_{50}$ = 192 µg/mL) compared to K562 cells (IC$_{50}$ = 500 µg/mL) [13]. However, K562 cells demonstrated increased apoptosis and cell cycle arrest in the G2/M phase [13]. *D. pentandra* was also reported to have anticancer and antiproliferative activity toward cancerous cells. Antiproliferative activity against MCF-7 cells was conducted using various types of *D. pentandra* extract: chloroform, diethyl ether, ethyl acetate, methanol, and petroleum ether. The ethyl acetate (IC$_{50}$ = 14.42 µg/mL) and methanolic (IC$_{50}$ = 17.70 µg/mL) extracts were found to be more effective against MCF-7 cancer cell lines compared to the other extracts. Compared to the normal cell L929, all extracts showed no remarkable effect with IC$_{50}$ values compared to 100 µg/mL and thus did not affect proliferation toward normal cells [15].

Furthermore, *D. pentandra* ethanolic extracts have been shown to enhance the anticancer effect of doxorubicin in MCF-7 cells [17]. The intracellular calcium and survivin levels were reduced significantly, while the combination of apoptotic cells with doxorubicin (5 µg/mL) and *D. pentandra* extracts were investigated (25 and 50 µg/mL) [17]. In addition, the combination of 5-fluorouracil (5 µg/mL) and ethanolic extracts (50 µg/mL) of *D. pentandra* was able to increase the apoptosis cells and p21 expression and decrease the survivin expression of HeLa cells [18]. Therefore, *D. pentandra* extracts are a promising

candidate in cancer therapeutics and have the potential to be developed as a new cancer drug in the future.

### 5.4. Antidiabetic and Antihyperglycaemic Activities

The use of plant-derived drugs for diabetes treatment has been approved by the World Health Organization due to the fact that medicinal plants are non-toxic, lack side effects compared to allopathic medicine, and have traditionally proven their safety [78]. Methanolic and water extracts of *D. pentandra* from various hosts show significant antidiabetic activity. From the $\alpha$-glucosidase inhibitor assay, IC$_{50}$ ranged between 11.8 and 50.9 $\mu$g/mL for all the species tested [1]. Overall, water extracts had higher antidiabetic activity as lower IC$_{50}$ were recorded compared to the methanolic extracts [1]. The highest antidiabetic activity was from the water extract of *D. pentandra* grown on *Camellia sinensis* (IC$_{50}$ = 11.8 $\mu$g/mL) [1].

*D. pentandra* also possessed antihyperglycaemic asides from antidiabetic activity. Single doses (400 mg/kg BW) of *D. pentandra* methanolic extracts displayed significant antidiabetic activity compared to the activity of the standard antihyperglycaemic agent metformin [19]. The flavonoid content of the methanolic leaf extract of *D. pentandra* was primarily responsible for its antidiabetic and antihyperglycemic properties [19].

### 5.5. Other Biological Activities

The LC$_{50}$ (lethal concentration 50) values of the extracts were used for the cytotoxicity effect. Based on the Artanti et al. [1] study, the aqueous and methanolic extracts of *D. pentandra* were non-toxic (LC$_{50}$ > 1000 $\mu$g/mL). Yee et al. [16] also reported the same results for the *D. pentandra* ethyl acetate extract (LC$_{50}$ > 1000 ppm). Nonetheless, Elsyana et al. [14] reported that the hexane fraction of *D. pentandra* exhibited significant cytotoxicity with an LC$_{50}$ value of 55.32 $\mu$g/mL.

The methanolic extract of *D. pentandra* was reported to have hepatoprotective activity. The extracts at a dose of 400 mg/kg BW significantly lowered liver damage parameters such as total cholesterol, total protein, serum alanine aminotransferase, and aspartate aminotransferase when administered to paracetamol-treated rats [19]. The hepatoprotective activity was contributed by quercetin [19].

Moreover, *D. pentandra* was also reported to have anti-inflammatory activity. The ethanolic extracts of *D. pentandra* that grew on mango potently suppressed colon shortening and myeloperoxidase in mice with 2,4,6-trinitrobenzene sulfonic acid-induced colitis [20]. The extracts at 600 mg/kg BW could inhibit Th17 cell differentiation by inhibiting IL-17 production while increasing the production of Treg-associated cytokines in the mesenteric lymph node cells [20]. It was suggested that the extracts containing quercetin played an essential role in inhibiting inflammation.

In addition to that, methanolic extracts of *D. pentandra* also possessed immunomodulatory activity. The extract stimulated the proliferation of mice splenocytes and thymocytes in a time- and dose-dependent manner [21]. Splenocytes from mice treated with 100 $\mu$g/mL of *D. pentandra* extract exhibited the highest proliferation [21]. The capability of these plant extracts to modulate innate immune functions through the modulation of lymphocytes suggested promising future therapeutic developments for wound healing and the inhibition of tumour growth [21].

Recently, the ethanolic extracts of *D. pentandra* were reported to have anti-aging activity. The ethanolic extract of *D. pentandra* leaves, which grew on the mango host, had excellent potential to inhibit the immunosenescence process in the immune system as characterised by a decrease in CD4$^+$ CD28$^-$ and CD8$^+$ CD28$^-$ and an increase in IL-2 levels [22].

## 6. Future Perspectives

Despite being commonly regarded as an undesirable plant due to its parasitic nature, *D. pentandra* can be recognised as a medicinal plant due to its health-promoting properties. According to Kwanda et al. [79], it is traditionally used to treat chronic diseases, high blood pressure, and immunological disorders. For instance, *V. album* (or European mistletoe) is the

most commonly consumed adjuvant among European cancer patients [80]. Additionally, traditional folklore has used mistletoes for bone dislocations and fractures, migraines, muscle swelling and sprains, postpartum, suspected cancer, and other ailments, even though the pharmacological effects of these mistletoes are currently understudied [81].

In a nutshell, research needs to be conducted more intensively on this plant by applying modern techniques such as SFE and UAE; therefore, the potential of drug raw materials from this plant can be further developed. The compounds identified and isolated from these plants are indispensable for investigating the biological activities of their extract, which are nature-provided hints regarding their potential use in medicine. As similar genera are assumed to have similar effects, extrapolating the results from other species within these genera could aid in the identification and study of various pharmacological effects. Identifying and quantifying the therapeutic uses of these extracts would pave the way for clinical studies if constituent-based research and documentation are conducted. Therefore, high-quality human-beneficial research is urgently required for these plants.

**Author Contributions:** Conceptualisation, M.A.A., N.N.N.N.M.D., N.I.M.I. and F.I.A.; methodology, M.A.Z.B.; software, M.A.Z.B.; validation, M.A.A.; formal analysis, M.A.Z.B.; investigation, M.A.A. and M.A.Z.B.; resources, M.A.A., N.N.N.N.M.D., N.I.M.I. and F.I.A.; data curation, M.A.A., N.N.N.N.M.D., N.I.M.I. and F.I.A.; writing—original draft preparation, M.A.A., N.N.N.N.M.D., N.I.M.I. and F.I.A.; writing—review and editing, M.A.A. and M.A.Z.B.; visualisation, M.A.Z.B.; supervision, M.A.A.; project administration, M.A.A.; funding acquisition, M.A.A. All authors have read and agreed to the published version of the manuscript.

**Funding:** This research received no external funding. The APC was funded by UMS Press, Universiti Malaysia Sabah.

**Data Availability Statement:** Not applicable.

**Acknowledgments:** The authors thank the reviewers and the academic editor whose comments helped improve this article.

**Conflicts of Interest:** The authors declare no conflict of interest.

## Abbreviations

| | |
|---|---|
| ASE | Accelerated solvent extraction |
| $CO_2$ | Carbon dioxide |
| DPPH | 2,2-Diphenyl-1-picrylhydrazyl |
| $IC_{50}$ | Half maximal inhibitory concentration |
| $LC_{50}$ | Lethal concentration 50 |
| MAE | Microwave-assisted extraction |
| miRNAs | microRNAs |
| ROS | Reactive oxygen species |
| SFE | Supercritical fluid extraction |
| UAE | Ultrasonic-assisted extraction |
| ZOI | Zone of inhibition |

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
