# Peer review of "A Review of Dendrophthoe pentandra (Mistletoe): Phytomorphology, Extraction Techniques, Phytochemicals, and Biological Activities"

_processes, doi:10.3390/pr11082348_

Round 1

Author Response

Dear Reviewer,

Thank you for your useful comments and suggestions on the structure of our manuscript. Please see the attachment for your perusal.

Thank you.

Reviewer 2 Report

The review article authored by Awang et al. sheds light on the medicinal significance of Dendrophtoe pentandra, a semi-parasitic plant indigenous to Southeast Asia that thrives on various host plants. The extracts derived from this plant have been extensively utilized in traditional and alternative medicine for the treatment of ailments such as coughs, diabetes, hypertension, and cancer. Extensive scientific investigations have further revealed its remarkable potential for various medicinal properties, including antioxidant, antibacterial, anticancer, antidiabetic, anti-inflammatory, hepatoprotective, immunomodulatory, and anti-aging effects. The authors also delve into different extraction techniques employed to isolate and identify a plethora of phytochemical compounds from almost every part of the plant. Overall, this article provides valuable insights into the promise of medicinal plants as a potential solution to the escalating challenges posed by disease and drug resistance.

Merit

1. The article is well-organized and provides a comprehensive review of the current literature on the topic.

2. Medicinal plants hold immense significance in the treatment of life-threatening diseases, providing valuable therapeutic options and serving as a vital resource for modern medicine. Among these plants, Dendrophtoe pentandra stands out for its remarkable importance and diverse applications in combating a range of severe illnesses.

These are the minor concern to improve the manuscript.

11.  Figure numbers are incorrectly cited in the manuscript.

22. The manuscript lacks a clear and thorough explanation of the extraction techniques employed. It would greatly benefit the broader research community if the authors could provide a pictorial representation illustrating the different extraction techniques and delve into greater detail about each method. This would enhance the understanding of the extraction processes and make the information more accessible and applicable to researchers in the field.

Author Response

(The authors gave the same response as above.)
